# A Primer on Multi-Neuron Relaxation-based Adversarial Robustness Certification

**Kevin Roth** [1]

## Abstract

The existence of adversarial examples poses a real danger when deep neural networks are deployed in the real world. The go-to strategy to quantify this vulnerability is to evaluate the model against specific attack algorithms. This approach is however inherently limited, as it says little about the robustness of the model against more powerful attacks not included in the evaluation. We develop a unified mathematical framework to describe relaxation-based robustness certification methods, which go beyond adversary-specific robustness evaluation and instead provide provable robustness guarantees against attacks by any adversary. We discuss the fundamental limitations posed by single-neuron relaxations and show how the recent "k-ReLU" multi-neuron relaxation framework of (Singh et al., 2019a) obtains tighter correlation-aware activation bounds by leveraging additional relational constraints among groups of neurons. Specifically, we show how additional pre-activation bounds can be mapped to corresponding post-activation bounds and how they can in turn be used to obtain tighter robustness certificates. We also present an intuitive way to visualize different relaxation-based certification methods. By approximating multiple non-linearities jointly instead of separately, the k-ReLU method is able to bypass the convex barrier imposed by single neuron relaxations. Full version: https://arxiv.org/abs/2106.03099

## 1. Introduction

**Adversarial Examples** While deep neural networks have been used with great success for perceptual tasks such as image classification or speech recognition, their performance can deteriorate dramatically in the face of so-called adver-

[1]Department of Computer Science, ETH Zürich. Correspondence to: Kevin Roth <kevin.roth@inf.ethz.ch>.

*Accepted by the ICML 2021 workshop on A Blessing in Disguise: The Prospects and Perils of Adversarial Machine Learning.* Copyright 2021 by the author(s).

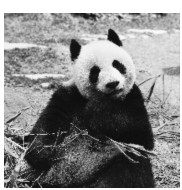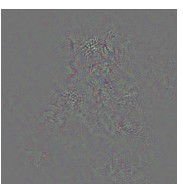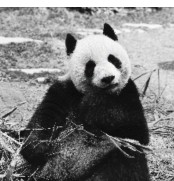

*Figure 1.* Adversarial Examples: (left) original image "giant panda" (ImageNet class 339), (middle) adversarial perturbation, (right) adversarially perturbed image causing misclassification "goldfish" (ImageNet class 2) on a MobileNetV2 pretrained on ImageNet.

sarial examples (Biggio et al., 2013; Szegedy et al., 2013; Goodfellow et al., 2014), i.e. small specifically crafted perturbations of the input signal, often imperceptible to humans, that are sufficient to induce large changes in the model output, cf. Figure 1. Ever since their discovery, there has been a huge interest in the machine learning community to try to understand their origin and to mitigate their consequences.

**Robustness Certification** The existence of adversarial examples poses a real danger when deep neural networks are deployed in the real world. The go-to strategy to quantify this vulnerability is to evaluate the model against specific attack algorithms. This approach is however inherently limited, as it says little about the robustness of the model against more powerful attacks not included in the evaluation (Carlini & Wagner, 2017; Athalye et al., 2018). We therefore need to go beyond adversary-specific robustness evaluation and instead provide provable robustness guarantees against attacks by any adversary.

The idea of robustness certification is to find the largest neighborhood (typically norm-bounded) that guarantees that no perturbation inside it can change the network's prediction, or equivalently to find the minimum distortion required to induce a misprediction. Unfortunately, exactly certifying the robustness of a network is an NP-complete problem (Katz et al., 2017; Weng et al., 2018). Consider a ReLU network: to find the exact mimimum distortion, two branches have to be considered for each ReLU activation where the input can take both positive and negative values. This makes exact verification methods computationally demanding even for small networks (Katz et al., 2017; Weng et al., 2018; Zhang et al., 2018).

**Relaxation-based certification** While exact certification is hard, providing a guaranteed certified lower bound for the minimum adversarial perturbation, resp. an upper bound on the robust error (i.e. the probability of misprediction for a given uncertainty set), can be done efficiently with relaxation-based certification methods (Hein & Andriushchenko, 2017; Wong & Kolter, 2018; Raghunathan et al., 2018; Dvijotham et al., 2018; Weng et al., 2018; Zhang et al., 2018; Singh et al., 2018; Mirman et al., 2018; Singh et al., 2019b; Qin et al., 2019; Salman et al., 2019). Relaxation-based verifiers trade off precision (increased false negative rates) with efficiency and scalability by convexly relaxing the non-linearities in the network. The resulting certificates are still sound: they may fail to verify robustness for a data point that is actually robust, but they never falsely certify a data point that is not robust.

**Single-Neuron Relaxation Barrier** The effectiveness of existing single-neuron relaxation-based verifiers is inherently limited by the optimal convex relaxation obtainable by processing each non-linearity separately (Salman et al., 2019). Relaxing the non-linearities separately comes at the cost of losing correlations between neurons: While existing frameworks do consider correlations between units to compute bounds for higher layers, the bounds for neurons in a given layer are computed individually, i.e. without interactions within the same layer (Salman et al., 2019). Moreover, as the activation bounds are obtained recursively, there is a risk that the error amplifies across layers, which is particularly problematic for deep neural networks.

**Multi-Neuron Relaxation** The single-neuron relaxation barrier can be bypassed by considering multi-neuron relaxations, i.e. by approximating multiple non-linearities *jointly* instead of separately, as suggested in the recent LP-solver based "k-ReLU" relaxation framework of (Singh et al., 2019a). The "k-ReLU" relaxation framework obtains tighter activation bounds by leveraging additional relational constraints among groups of neurons, enabling significantly more precise certification than existing state-of-the-art verifiers while still maintaining scalability (Singh et al., 2019a). Finally, note that there is another conceptually different approach to bypass the single-neuron relaxation barrier: (Tjandraatmadja et al., 2020) obtain tightened single neuron relaxations by considering bounds in terms of the multivariate post-activations preceding the affine layer.

## 2. Single-Neuron Relaxation

**Notation.** Let $f : \mathcal{X} \subset \mathbb{R}^{n_0} \to \mathbb{R}^{n_L}$ be an $L$-layer deep feedforward neural network, given by the equations[1]

$$\mathbf{x}^{(i)} = \mathbf{W}^{(i)}\sigma(\mathbf{x}^{(i-1)}) + \mathbf{b}^{(i)} \quad \text{for} \quad i \in [\![L]\!], \quad (1)$$

with input $\mathbf{x}^{(0)} = \mathbf{x}$ and output $\mathbf{x}^{(L)} \equiv f(\mathbf{x})$, where $\mathbf{W}^{(i)}, \mathbf{b}^{(i)}$ denote the layer-wise weight matrix and bias vector, and where $\sigma(\cdot)$ denotes the non-linear activation

function, with the convention that $\sigma(\mathbf{x}^{(0)}) = \mathbf{x}$ is the identity activation. We use $n_i := \dim(\mathbf{x}^{(i)})$ to denote the number of neurons in layer $i$, and $\mathbf{x}^{[\![L_0, L_1]\!]} := \{\mathbf{x}^{(L_0)}, \dots, \mathbf{x}^{(L_1)}\}$ to denote the collection of all $\mathbf{x}^{(i)}$ for $i \in [\![L_0, L_1]\!]$, where $[\![L_0, L_1]\!]$ denotes the set of indices $[\![L_0, L_1]\!] := \{L_0, \dots, L_1\}$, with the shorthand notation $[\![L]\!] \equiv [\![1, L]\!]$. Note that in our notation the pre-activations $\mathbf{x}^{(i)}$ have the same index as the weight matrix $\mathbf{W}^{(i)}$ and bias vector $\mathbf{b}^{(i)}$ on which they directly depend. Together, the above equations[1] define the mapping $f(\cdot) = \mathbf{W}^{(L)}\sigma(\cdots \mathbf{W}^{(2)}\sigma(\mathbf{W}^{(1)}(\cdot) + \mathbf{b}^{(1)}) + \mathbf{b}^{(2)} \cdots) + \mathbf{b}^{(L)}$.

**Robustness Certification.** The network is considered certifiably robust with respect to input $\mathbf{x}$ and uncertainty set $\mathcal{B}(\mathbf{x})$ if there is no perturbation within $\mathcal{B}(\mathbf{x})$ that can change the network's prediction, i.e. if the largest logit $f_{\hat{k}(\mathbf{x})}$ remains larger than any other logit $f_k$ within the entire uncertainty set $\mathcal{B}(\mathbf{x})$. Formally, the network is *certifiably robust* at $\mathbf{x}$ with respect to $\mathcal{B}(\mathbf{x})$ if

$$\min_{\mathbf{x}^* \in \mathcal{B}(\mathbf{x})} f_{\hat{k}(\mathbf{x})}(\mathbf{x}^*) - f_k(\mathbf{x}^*) > 0, \quad \forall k \neq \hat{k}(\mathbf{x}), \quad (2)$$

where $\hat{k}(\mathbf{x}) = \arg\max_j f_j(\mathbf{x})$.

Typical choices for $\mathcal{B}(\mathbf{x})$ are the $\ell_p$-norm balls $\mathcal{B}_p(\mathbf{x}; \epsilon) := \{\mathbf{x}^* : ||\mathbf{x}^* - \mathbf{x}||_p \leq \epsilon\}$ or the non-uniform box domain $\mathcal{B}(\mathbf{x}; \{\epsilon_j^-, \epsilon_j^+\}_{j \in [\![n_0]\!]}) := \{\mathbf{x}^* : \mathbf{x}_j - \epsilon_j^- \leq \mathbf{x}_j^* \leq \mathbf{x}_j + \epsilon_j^+\}$ with $\epsilon_j^-, \epsilon_j^+ \in \mathbb{R}_0^+$.

The optimization domain in the above certification problem can be narrowed down significantly if we are given lower- and upper- *pre-activation bounds*, $\boldsymbol{\ell}^{(i)} \leq \mathbf{x}^{(i)} \leq \boldsymbol{u}^{(i)}$, for $i \in [\![L-1]\!]$. We will see below how (approximate) pre-activation bounds can be computed. The corresponding optimization problem, subsequently referred to via the shorthand notation $\mathcal{O}(\mathbf{c}, c_0, L, \boldsymbol{\ell}^{[\![L-1]\!]}, \boldsymbol{u}^{[\![L-1]\!]})$, reads

$$\min_{\mathbf{x}^{[\![0,L]\!]}} \mathbf{c}^\top \mathbf{x}^{(L)} + c_0 \qquad\qquad (3)$$
$$\text{s.t. } \mathbf{x}^{(i)} = \mathbf{W}^{(i)}\sigma(\mathbf{x}^{(i-1)}) + \mathbf{b}^{(i)} \text{ for } i \in [\![L]\!]$$
$$\mathbf{x}^{(0)} \in \mathcal{B}(\mathbf{x}), \, \boldsymbol{\ell}^{(i)} \leq \mathbf{x}^{(i)} \leq \boldsymbol{u}^{(i)} \text{ for } i \in [\![L-1]\!]$$

where the vector-inequalities are considered to hold element-wise, i.e. $\boldsymbol{\ell}_j^{(i)} \leq \mathbf{x}_j^{(i)} \leq \boldsymbol{u}_j^{(i)}, \forall j \in [\![n_i]\!]$. For $\mathbf{c} \in \{\mathbf{e}_{\hat{k}(\mathbf{x})} - \mathbf{e}_k\}_{k \neq \hat{k}(\mathbf{x})}$, $c_0 = 0$ and $\boldsymbol{\ell}_j^{(i)} = -\infty, \boldsymbol{u}_j^{(i)} = \infty$, this formulation is equivalent to Equation 2.

We denote the optimal value of $\mathcal{O}(\mathbf{c}, c_0, L, \boldsymbol{\ell}^{[\![L-1]\!]}, \boldsymbol{u}^{[\![L-1]\!]})$ by $p_{\mathcal{O}}^*$. If $p_{\mathcal{O}}^* > 0$ for all $\mathbf{c} \in \{\mathbf{e}_{\hat{k}(\mathbf{x})} - \mathbf{e}_k\}_{k \neq \hat{k}(\mathbf{x})}$, with $c_0 = 0$ and valid pre-activation bounds $\boldsymbol{\ell}^{[\![L-1]\!]}, \boldsymbol{u}^{[\![L-1]\!]}$, the network is certifiably robust with respect to $\mathbf{x}$ and $\mathcal{B}(\mathbf{x})$.

---

[1]This formulation captures all network architectures in which units in one layer receive inputs from units in the previous layer, including fully-connected and convolutional networks. To capture residual networks, in which units receive inputs from units in multiple previous layers, the right hand side of the equation would have to be extended to include dependencies on $\mathbf{x}^{(j)}$ for $j < i-1$.

Exact pre-activation bounds $\boldsymbol{\ell}_j^{(i)}$ and $\boldsymbol{u}_j^{(i)}$ are given by the minimization $(+\mathbf{e}_j)$ resp. maximization $(-\mathbf{e}_j)$ problems $\mathcal{O}(\pm\mathbf{e}_j, 0, i, \boldsymbol{\ell}^{[\![i-1]\!]}, \boldsymbol{u}^{[\![i-1]\!]})$ for all neurons $j \in [\![n_i]\!]$ and layers $i \in [\![L-1]\!]$. Unfortunately, computing exact bounds is as NP-hard as solving the exact certification problem.

Most of the existing robustness certification methods in use today are based on a variant of one of two (pre)-activation bound computation paradigms: (i) Interval Bound Propagation or (ii) Relaxation-based Bound Computation. The pre-activation bounds play a crucial role in these certification methods: the tighter they are, the lower the false-negative rate of the method.

**Relaxation-based Certification** Exactly certifying the robustness of a network via Equation 3 is an NP-complete problem (Katz et al., 2017; Weng et al., 2018), due to the non-convex constraints imposed by the non-linearities. Relaxation-based verifiers trade off precision with efficiency and scalability by convexly relaxing (over-approximating) the non-linearities in the network.

The convex relaxation-based certification problem, subsequently referred to via the short-hand notation $\mathcal{C}(\mathbf{c}, c_0, L, \boldsymbol{\ell}^{[\![L-1]\!]}, \boldsymbol{u}^{[\![L-1]\!]})$, reads

$$\min_{\mathbf{x}^{[\![0,L]\!]}, \, \mathbf{z}^{[\![0,L-1]\!]}} \quad \mathbf{c}^\top \mathbf{x}^{(L)} + c_0 \qquad (4)$$

$$\text{s.t.} \quad \mathbf{x}^{(i)} = \mathbf{W}^{(i)} \mathbf{z}^{(i-1)} + \mathbf{b}^{(i)} \text{ for } i \in [\![L]\!]$$

$$\underline{\sigma}(\mathbf{x}^{(i)}) \leqslant \mathbf{z}^{(i)} \leqslant \overline{\sigma}(\mathbf{x}^{(i)}) \text{ for } i \in [\![L-1]\!]$$

$$\mathbf{x}^{(0)} \in \mathcal{B}(\mathbf{x}), \, \boldsymbol{\ell}^{(i)} \leqslant \mathbf{x}^{(i)} \leqslant \boldsymbol{u}^{(i)} \text{ for } i \in [\![L-1]\!]$$

where $\underline{\sigma}(\mathbf{x}^{(i)})$ resp. $\overline{\sigma}(\mathbf{x}^{(i)})$ are convex resp. concave bounding functions satisfying $\underline{\sigma}(\mathbf{x}^{(i)}) \leqslant \sigma(\mathbf{x}^{(i)}) \leqslant \overline{\sigma}(\mathbf{x}^{(i)})$ for all $\boldsymbol{\ell}^{(i)} \leqslant \mathbf{x}^{(i)} \leqslant \boldsymbol{u}^{(i)}$, and where $\mathbf{z}^{(i)}$ are the post-activation variables with the convention that $\mathbf{z}^{(0)} = \mathbf{x}^{(0)}$. We denote the optimal value of $\mathcal{C}(\mathbf{c}, c_0, L, \boldsymbol{\ell}^{[\![L-1]\!]}, \boldsymbol{u}^{[\![L-1]\!]})$ by $p_{\mathcal{C}}^*$. Naturally, we have that $p_{\mathcal{C}}^* \leqslant p_{\mathcal{O}}^*$. Thus, if $p_{\mathcal{C}}^* > 0$ for all $\mathbf{c} \in \{\mathbf{e}_{\hat{k}(\mathbf{x})} - \mathbf{e}_k\}_{k \neq \hat{k}(\mathbf{x})}$, with $c_0 = 0$ and valid pre-activation bounds $\boldsymbol{\ell}^{[\![L-1]\!]}, \boldsymbol{u}^{[\![L-1]\!]}$, the network is certifiably robust w.r.t. $\mathbf{x}$ and $\mathcal{B}(\mathbf{x})$.

**More on single-neuron relaxation** can be found in **Section 5.3 in the Appendix**.

**Single-Neuron Relaxation Barrier** The effectiveness of existing single-neuron relaxation-based verifiers is inherently limited by the tightness of the optimal single-neuron relaxation, defined in Section 5.3 in the Appendix. In a series of highly compute-intense experiments, (Salman et al., 2019) found that optimal single-neuron relaxation based verification, i.e. solving the certification problems $\mathcal{C}_{\text{opt}}(\pm\mathbf{e}_j, 0, i, \boldsymbol{\ell}^{[\![i-1]\!]}, \boldsymbol{u}^{[\![i-1]\!]})$ for all neurons $j \in [\![n_i]\!]$ and layers $i$, does not significantly improve upon the gap between verifiers that greedily compute approximate pre-

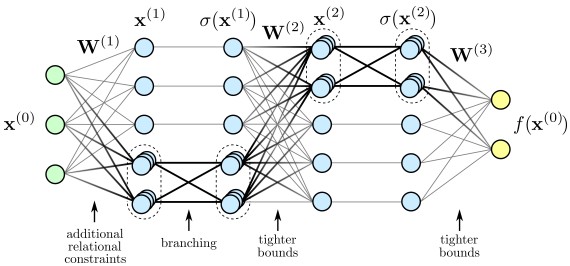

*Figure 2.* Multi-neuron relaxations obtain tighter *correlation-aware* activation bounds by leveraging additional relational constraints among groups of neurons, thereby overcoming the convex barrier imposed by the single-neuron relaxations.

activation bounds and only solve the relaxed (primal or dual) certification problem $\mathcal{C}_{\text{opt}}(\mathbf{c}, 0, L, \boldsymbol{\ell}^{[\![L-1]\!]}, \boldsymbol{u}^{[\![L-1]\!]})$ with $\mathbf{c} \in \{\mathbf{e}_{\hat{k}(\mathbf{x})} - \mathbf{e}_k\}_{k \neq \hat{k}(\mathbf{x})}$ for the logit layer, and the exact mixed integer linear programming (MILP) verifier from (Tjeng et al., 2018), suggesting that there is an inherent *barrier* to tight verification for single neuron relaxations. To improve the tightness of relaxation-based certification methods we therefore have to go beyond single-neuron relaxations.

# 3. Multi-Neuron Relaxation

The single-neuron relaxation barrier can be bypassed by considering multi-neuron relaxations, i.e. by approximating multiple non-linearities *jointly* instead of separately, as suggested in the recent LP-solver based "k-ReLU" relaxation framework of (Singh et al., 2019a). Singh et al.'s multi-neuron relaxation framework is specific to networks with ReLU non-linearities but can otherwise be incorporated into any certification method (primal or dual) that operates with pre-activation bounds, including (Weng et al., 2018; Zhang et al., 2018; Wong & Kolter, 2018).

In this Section, we show how the "k-ReLU" relaxation framework obtains tighter activation bounds by leveraging additional relational constraints among groups of neurons. In particular, we show how additional pre-activation bounds can be mapped to corresponding post-activation bounds and how they can in turn be used to obtain tighter bounds in higher layers, as illustrated in Figure 2. By capturing interactions between neurons, the k-ReLU method is able to overcome the convex barrier imposed by the single neuron relaxation (Salman et al., 2019).

Singh et al.'s experimental results indicate that k-ReLU enables significantly more precise certification than existing state-of-the-art verifiers while maintaining scalability. To illustrate the precision gain, Singh et al. measure the volume of the output bounding box computed after propagating an $\ell_\infty$-ball of radius $\epsilon = 0.015$ through a fully connected network with 9 layers containing 200 neurons each. They find

that the volume of the output from 2-ReLU resp. 3-ReLU relaxation is 7 resp. 9 orders of magnitude smaller than from single-neuron relaxation-based DeepPoly verifier (Singh et al., 2019a). Similarly, on the $9 \times 200$ fully-connected network resp. a convolutional network, k-ReLU certifies 506 resp. 347 adversarial regions whereas the single-neuron relaxation based RefineZono verifier certifies 316 resp. 179 adversarial regions (Singh et al., 2019a).

**ReLU branch polytopes** Following (Singh et al., 2019a), we consider the pre-activations $\mathbf{x}^{(i)}$ and the post-activations $\mathbf{z}^{(i)}$ as separate neurons. Let $\mathcal{S}^{(\ell)}$ be a convex set computed via some relaxation based certification method approximating the set of values that neurons $(\mathbf{x}^{[0,\ell]}, \mathbf{z}^{[0,\ell-1]})$, including the pre-activations $\mathbf{x}^{(\ell)}$ but excluding the post-activations $\mathbf{z}^{(\ell)}$, can take with respect to $\mathcal{B}(\mathbf{x})$,

$$
\begin{aligned}
\mathcal{S}^{(\ell)} := \Big\{ (\mathbf{x}^{[0,L]}, \mathbf{z}^{[0,L-1]}) \; : \qquad\qquad\qquad & (5) \\
\mathbf{x}^{(i)} = \mathbf{W}^{(i)} \mathbf{z}^{(i-1)} + \mathbf{b}^{(i)} \text{ for } i \in [\![\ell]\!] & \\
\underline{\sigma}(\mathbf{x}^{(i)}) \leqslant \mathbf{z}^{(i)} \leqslant \overline{\sigma}(\mathbf{x}^{(i)}) \text{ for } i \in [\![\ell-1]\!] & \\
\mathbf{x}^{(0)} \in \mathcal{B}(\mathbf{x}), \; \boldsymbol{\ell}^{(i)} \leqslant \mathbf{x}^{(i)} \leqslant \boldsymbol{u}^{(i)} \text{ for } i \in [\![\ell-1]\!] \Big\} &
\end{aligned}
$$

Note that, variables $(\mathbf{x}^{[\ell+1,L]}, \mathbf{z}^{[\ell,L-1]})$ that don't appear in the constraints, are considered to be *unconstrained*, i.e. they take values in the entire real number line.

In general, pre-activations $\mathbf{x}_j^{(\ell)}$ can take both positive and negative values in $\mathcal{S}^{(\ell)}$. For each ReLU activation where the input, i.e. the corresponding pre-activation, can take both positive and negative values, two branches have to be considered. Define the convex polytopes induced by the two branches of the $j$-th ReLU unit at layer $\ell$ as

$$
\begin{aligned}
C_{j+}^{(\ell)} &:= \{ (\mathbf{x}^{[0,L]}, \mathbf{z}^{[0,L-1]}) : \mathbf{x}_j^{(\ell)} \geqslant 0, \; \mathbf{z}_j^{(\ell)} = \mathbf{x}_j^{(\ell)} \} \\
C_{j-}^{(\ell)} &:= \{ (\mathbf{x}^{[0,L]}, \mathbf{z}^{[0,L-1]}) : \mathbf{x}_j^{(\ell)} \leqslant 0, \; \mathbf{z}_j^{(\ell)} = 0 \} ,
\end{aligned} \quad (6)
$$

which can be written more concisely as

$$
C_{j\,s_j}^{(\ell)} := \Big\{ (\mathbf{x}^{[0,L]}, \mathbf{z}^{[0,L-1]}) : s_j \mathbf{x}_j^{(\ell)} \geqslant 0, \; \mathbf{z}_j^{(\ell)} = \frac{1+s_j}{2} \mathbf{x}_j^{(\ell)} \Big\}.
\tag{7}
$$

Next, we introduce some notation to describe the polytopes representing the possible ways of selecting one ReLU branch per neuron. Let $J \subseteq [\![n_\ell]\!]$ be some index set over neurons, with cardinality $|J|$. For a specific configuration $(s_{j_1}, \ldots, s_{j_{|J|}}) \in \{+,-\}^{|J|}$ of individual ReLU branches, out of all $2^{|J|}$ possible configurations, let

$$
Q_{J,(s_{j_1}, \ldots, s_{j_{|J|}})}^{(\ell)} := \bigcap_{j \in J} C_{j\,s_j}^{(\ell)}
\tag{8}
$$

be the convex polytope defined as the intersection of the $|J|$ individual ReLU branch polytopes $\{C_{j\,s_j}^{(\ell)}\}_{j \in J}$.

Next, define $Q_J^{(\ell)}$ as the collection of all $2^{|J|}$ convex polytopes $Q_{J,(s_{j_1}, \ldots, s_{j_{|J|}})}^{(\ell)}$, indexed by all possible ReLU branch configurations $(s_{j_1}, \ldots, s_{j_{|J|}}) \in \{+,-\}^{|J|}$,

$$
Q_J^{(\ell)} := \Big\{ \bigcap_{j \in J} C_{j\,s_j}^{(\ell)} \; \Big| \; (s_{j_1}, \ldots, s_{j_{|J|}}) \in \{+,-\}^{|J|} \Big\}.
\tag{9}
$$

**Additional Relational Constraints** The k-ReLU relaxation framework bypasses the single-neuron convex barrier by leveraging additional relational constraints among groups of neurons. Specifically, the k-ReLU framework computes bounds on additional relational constraints of the form $\sum_{j \in J} a_j \mathbf{x}_j^{(\ell)}$. These additional relational constraints, together with the usual interval bounds, are captured by the convex polytope $P_{J,\mathcal{A}}^{(\ell)}$. Formally, let $P_{J,\mathcal{A}}^{(\ell)} \supseteq \mathcal{S}^{(\ell)}$ be a convex polytope containing interval constraints (for $(a_{j_1}, \ldots, a_{j_{|J|}})$ with $a_j \in \{-1,0,1\}$, $\sum_{j \in J} |a_j| = 1$) and relational constraints (for general $(a_{j_1}, \ldots, a_{j_{|J|}})$, with $a_j \in \mathbb{R}$) over neurons $\mathbf{x}_j^{(\ell)}, j \in J$, defined as

$$
\begin{aligned}
P_{J,\mathcal{A}}^{(\ell)} := \Big\{ (\mathbf{x}^{[0,L]}, \mathbf{z}^{[0,L-1]}) : \sum_{j \in J} a_j \mathbf{x}_j^{(\ell)} & \\
\leqslant c^{(\ell)}(a_{j_1}, \ldots, a_{j_{|J|}}) \Big| (a_{j_1}, \ldots, a_{j_{|J|}}) \in \mathcal{A} \Big\}, &
\end{aligned}
\tag{10}
$$

where the set $\mathcal{A}$ contains the coefficient-tuples of all the constraints defining $P_{J,\mathcal{A}}^{(\ell)}$.

In practice, the corresponding bounds $c^{(\ell)}(a_{j_1}, \ldots, a_{j_{|J|}})$ on interval and relational constraints over neurons $\mathbf{x}_j^{(\ell)}, j \in J$, can be computed using any of the existing bound computation algorithms, e.g. (Zhang et al., 2018), Algorithm 1 in (Wong & Kolter, 2018) or DeepPoly (Singh et al., 2019a). Using the notation for the relaxation-based certification problem in Equation 4, the bounds are given as

$$
c^{(\ell)}(a_{j_1}, \ldots, a_{j_{|J|}}) = \mathcal{C}(-\mathbf{a}, 0, \ell, \boldsymbol{\ell}^{[\ell-1]}, \boldsymbol{u}^{[\ell-1]})
\tag{11}
$$

with $\mathbf{a} = \sum_{j \in J} a_j \mathbf{e}_j$, where $\mathbf{e}_j$ denotes the $j$-th canonical basis vector.

Ideally, one would like $P_{J,\mathcal{A}}^{(\ell)}$ to be the projection of $\mathcal{S}^{(\ell)}$ onto the variables $\mathbf{x}_j^{(\ell)}$ indexed by $J$. However, computing this projection is prohibitively expensive. Singh et al. (2019a) heuristically found $\mathcal{A} = \{(a_{j_1}, \ldots, a_{j_{|J|}}) \in \{-1,0,1\}^{|J|} \backslash (0, \ldots, 0)\}$, containing $3^{|J|} - 1$ constraints ($2|J|$ interval and $3^{|J|} - 2|J| - 1$ relational), to work well in practice. It remains an open problem of whether there exists a theoretically optimal arrangement of a given number of additional relational constraints. Geometrically, $P_{J,\mathcal{A}}^{(\ell)}$ is an over-approximation of the projection of $\mathcal{S}^{(\ell)}$ onto the variables $\mathbf{x}_j^{(\ell)}$ indexed by $J$.

**Optimal Convex Multi-Neuron Relaxation** The optimal convex relaxation of the $n_\ell$ ReLU assignments considers all $n_\ell$ neurons jointly

$$\mathcal{S}_{\text{opt}}^{(\ell)} = \mathcal{CH}\Big( \bigcup_{Q \in Q_{\llbracket n_\ell \rrbracket}^{(\ell)}} \{\mathcal{S}^{(\ell)} \cap Q\} \Big), \qquad (12)$$

where $Q_{\llbracket n_\ell \rrbracket}^{(\ell)}$ is the collection of the $2^{n_\ell}$ convex polytopes $Q_{(s_1,\dots,s_{n_\ell})}^{(\ell)} := \bigcap_{j \in \llbracket n_\ell \rrbracket} C_{j\,s_j}^{(\ell)}$ indexed by all possible ReLU branch configurations across neurons at layer $\ell$, and where $\mathcal{CH}$ denotes the Convex-Hull.

**k-ReLU Relaxation** The k-ReLU framework partitions the $n_\ell$ neurons into $n_\ell/k$ disjoint sub-groups of size $k$, then jointly relaxes the neurons within each sub-group, as a compromise between the practically infeasible joint relaxation of all $n_\ell$ neurons and the single-neuron relaxation barrier arising when relaxing each neuron individually.

Suppose that $n_\ell$ is divisible by $k$. Let $\mathcal{J} = \{J_i\}_{i=1}^{n_\ell/k}$ be a partition of the set of neurons $\llbracket n_\ell \rrbracket$ such that each group $J_i \in \mathcal{J}$ contains exactly $|J_i| = k$ indices. (Singh et al., 2019a) k-ReLU framework computes the following convex relaxation

$$\mathcal{S}_{\text{k-ReLU}}^{(\ell)} = \mathcal{S}^{(\ell)} \cap \bigcap_{i=1}^{n_\ell/k} \mathcal{CH}\Big( \bigcup_{Q \in Q_{J_i}^{(\ell)}} \{P_{J_i, \mathcal{A}_i}^{(\ell)} \cap Q\} \Big). \quad (13)$$

Conceptually, each convex polytope $P_{J_i, \mathcal{A}_i}^{(\ell)}$ (containing interval and relational constraints on neurons $\mathbf{x}_j^{(\ell)}, j \in J_i$) is intersected with convex ReLU branch polytopes $Q$ from the set $Q_{J_i}^{(\ell)}$, producing $2^k$ convex polytopes $\{P_{J_i, \mathcal{A}_i}^{(\ell)} \cap Q\}_{Q \in Q_{J_i}^{(\ell)}}$, one for each possible branch $Q \in Q_{J_i}^{(\ell)}$.

The union $\bigcup_{Q \in Q_{J_i}^{(\ell)}} \{P_{J_i, \mathcal{A}_i}^{(\ell)} \cap Q\}$ of all $2^k$ convex polytopes captures[2] the uncertainty set over post-activations $\mathbf{z}_j^{(\ell)}$ indexed by $j \in J_i$. As the different groups of neurons are disjoint $J_i \cap J_j = \varnothing$, we can combine those group-specific post-activation uncertainty sets by intersecting their convex hulls $\mathcal{K}_i = \mathcal{CH}\big( \bigcup_{Q \in Q_{J_i}^{(\ell)}} \{P_{J_i, \mathcal{A}_i}^{(\ell)} \cap Q\} \big)$. From this, $\mathcal{S}_{\text{k-ReLU}}^{(\ell)}$ is obtained by intersection with the convex set $\mathcal{S}^{(\ell)}$.

(Singh et al., 2019a) heuristically chose the partition $\mathcal{J} = \{J_i\}_{i=1}^{n_\ell/k}$ such that neurons $j \in \llbracket n_\ell \rrbracket$ are grouped according to the area of their triangle relaxation (i.e. neurons with similar triangle relaxation areas are grouped together). It remains an open problem whether there exists a theoretically optimal partitioning of the groups of neurons.

---

[2]"over-approximates" (to be more precise), since $P_{J_i, \mathcal{A}_i}^{(\ell)}$ is an over-approximation of the projection of $\mathcal{S}^{(\ell)}$ onto the variables indexed by $J_i$.

**Convex Hull Computation** Singh et al. use the *cdd* library to compute convex hulls (cdd, 2021). *cdd* is an implementation of the double description method by Motzkin, that allows to compute all vertices (i.e. extreme points) of a general convex polytope given as a system of linear inequalities. cdd also implements the reverse operation, allowing to compute the convex hull from a set of vertices. In practice, *cdd* is used first to compute the vertices of the convex polytopes $\{P_{J_i, \mathcal{A}_i}^{(\ell)} \cap Q\}_{Q \in Q_{J_i}^{(\ell)}}$ and second to compute the convex hull of the union of the vertices of all these polytopes.

**Tighter bounds in higher layers** Finally, we show how the k-ReLU relaxation $\mathcal{S}_{\text{k-ReLU}}^{(\ell)}$ can be used to obtain tighter bounds in higher layers. The k-ReLU method computes *refined* pre-activation bounds $\boldsymbol{\ell}_j^{(i+1)}, \boldsymbol{u}_j^{(i+1)}$ for neurons at layer $\ell + 1$, by maximizing and minimizing $\mathbf{x}_j^{(i+1)} = \mathbf{W}_{j\cdot\cdot}^{(\ell+1)} \mathbf{z}^{(\ell)} + \mathbf{b}_j^{(\ell)}$ w.r.t. $\mathbf{z}^{(\ell)}$ subject to $\mathbf{z}^{(\ell)} \in \mathcal{S}_{\text{k-ReLU}}^{(\ell)}$,

$$\boldsymbol{\ell}_j^{(\ell+1)} = \min_{\mathbf{z}^{(\ell)} \in \mathcal{S}_{\text{k-ReLU}}^{(\ell)}} \mathbf{W}_{j\cdot\cdot}^{(\ell+1)} \mathbf{z}^{(\ell)} + \mathbf{b}_j^{(\ell+1)} \qquad (14)$$

$$\boldsymbol{u}_j^{(\ell+1)} = \max_{\mathbf{z}^{(\ell)} \in \mathcal{S}_{\text{k-ReLU}}^{(\ell)}} \mathbf{W}_{j\cdot\cdot}^{(\ell+1)} \mathbf{z}^{(\ell)} + \mathbf{b}_j^{(\ell+1)} \qquad (15)$$

Since all the constraints in $\mathcal{S}_{\text{k-ReLU}}^{(\ell)}$ are linear, we can use an LP-solver for the maximization and minimization. Singh et al. use the *gurobi* solver (Gurobi, 2021).

**Interpretation** Computing bounds on the additional relational constraints corresponds to solving a bounding problem on a widened network, that is equal to the original network up to layer $\ell - 1$ but with a *"modified"* $\ell$-th layer which includes additional neurons representing linear combinations of rows of the $\ell$-th layer weight matrix with coefficients determined by $(a_{j_1}, \dots, a_{j_{|J|}})$. See Figure 2.

## 4. Discussion

We have shown how the recent "k-ReLU" framework of (Singh et al., 2019a) obtains tighter correlation-aware activation bounds by leveraging additional relational constraints among groups of neurons. In particular, we have shown how additional pre-activation constraints can be mapped to corresponding post-activation constraints and how they can in turn be used to obtain tighter pre-activation bounds in higher layers. The k-ReLU framework is specific to ReLU networks but can otherwise be incorporated into any certification method that operates with pre-activation bounds.

The main degrees of freedom in k-ReLU are the partitioning of the neurons into sub-groups and the choice of coefficients $a_j$ in the additional relational constraints. We consider it to be an interesting avenue of research to investigate whether there is a theoretically optimal partitioning as well as choice for the coefficients of the additiona relational constraints.

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

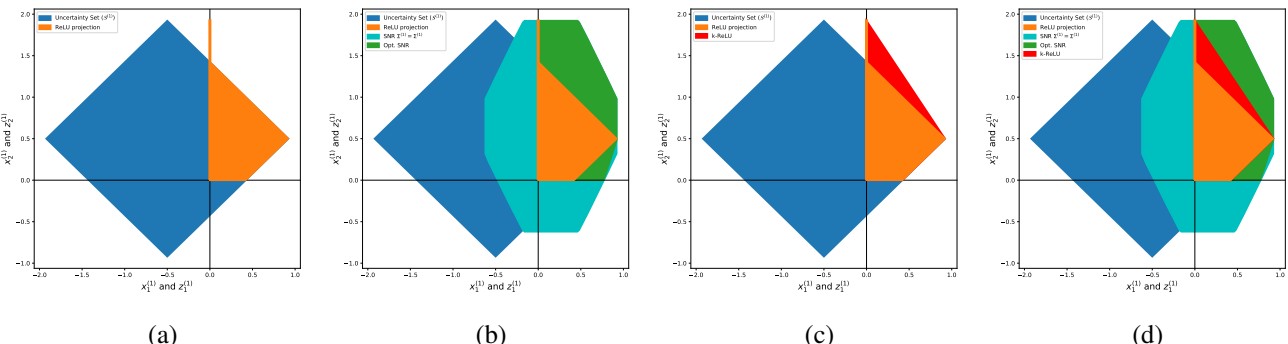

*Figure 3.* Visualization of different relaxation methods. (a) Uncertainty set after the first affine layer (i.e. $\mathcal{S}^{(1)}$ projected onto $\mathbf{x}^{(1)}$) and its ReLU projection, (b) uncertainty set, its ReLU projection, single-neuron relaxation (SNR) with same relaxation slope matrices $\underline{\boldsymbol{\Sigma}}^{(1)} = \overline{\boldsymbol{\Sigma}}^{(1)}$ and SNR with optimal single-neuron relaxation (note that the same slope SNR set contains the optimal SNR set which in turn contains the ReLU projection set), (c) uncertainty set, its ReLU projection and the k-ReLU relaxation (corresponding to the convex hull of the ReLU projection), (d) everything combined for comparison. Note that corresponding pre- and post-activation pairs are superimposed on the same axis, i.e. both $\mathbf{x}_1^{(1)}$ and $\mathbf{z}_1^{(1)}$ are plotted (superimposed) on the horizontal axis while both $\mathbf{x}_2^{(1)}$ and $\mathbf{z}_2^{(1)}$ are plotted (superimposed) on the vertical axis. This style of visualization, where pre- and post-activation pairs are superimposed, allows to easily convey the effect of the non-linearities.

# 5. Appendix

## 5.1. Visualization of Relaxation Methods

## 5.2. Notation

We introduce the following additional notation:

**Definition 1.** *(Row / column indexing). Let* $\mathbf{A} \in \mathbb{R}^{m \times n}$ *be an arbitrary real-valued matrix. We use* $\mathbf{A}_{i\cdot\cdot}$ *to denote the* $i$*-th row and* $\mathbf{A}_{\cdot j}$ *to denote the* $j$*-th column of* $\mathbf{A}$*, for* $i \in [\![m]\!]$ *and* $j \in [\![n]\!]$*.*

**Definition 2.** *(Positive / negative entries). Let* $\mathbf{A} \in \mathbb{R}^{m \times n}$ *be an arbitrary real-valued matrix. Define*

$$
\begin{aligned}
[\,\cdot\,]_+ : \mathbf{A} \rightarrow \mathbf{A}_+ &\coloneqq \max(\mathbf{A}, 0)\,, \\
[\,\cdot\,]_- : \mathbf{A} \rightarrow \mathbf{A}_- &\coloneqq \min(\mathbf{A}, 0)\,,
\end{aligned}
\tag{16}
$$

*where the* $\max(\cdot, \cdot)$ *and* $\min(\cdot, \cdot)$ *are taken* entrywise. *Note that by definition,* $\mathbf{A} = \mathbf{A}_+ + \mathbf{A}_-$*.*

**Definition 3.** *(Row-wise* $q$*-norm). Let* $\mathbf{A} \in \mathbb{R}^{m \times n}$ *be an arbitrary real-valued matrix. Define the row-wise* $q$*-norm* $||\cdot||_{q\cdot\cdot}$ *as the following column vector in* $(\mathbb{R}_0^+)^m$*:*

$$
||\mathbf{A}||_{q\cdot\cdot} = \left(||\mathbf{A}_{1\cdot\cdot}||_q, ||\mathbf{A}_{2\cdot\cdot}||_q, \ldots, ||\mathbf{A}_{m\cdot\cdot}||_q\right)^\top.
\tag{17}
$$

## 5.3. Single-Neuron Relaxation Continued

**Optimal Single-Neuron Relaxation**   Under mild assumptions (non-interactivity, see below), the optimal convex relaxation of a single non-linearity, i.e. its convex hull, is given by $\underline{\sigma}_{\mathrm{opt}}(x) \leqslant \sigma(x) \leqslant \overline{\sigma}_{\mathrm{opt}}(x)$ (Salman et al., 2019), where

$$
\begin{aligned}
\underline{\sigma}_{\mathrm{opt}}(x) &\text{ is the greatest convex function majored by } \sigma \\
\overline{\sigma}_{\mathrm{opt}}(x) &\text{ is the smallest concave function majoring } \sigma
\end{aligned}
\tag{18}
$$

In general, for vector-valued non-linearities $\sigma : \mathbb{R}^n \rightarrow \mathbb{R}^m$, the optimal convex relaxation may not have a simple analytic form. However, if there is no interaction among the outputs $\sigma_i$ for $i \in [\![m]\!]$, the optimal convex relaxation does admit a simple analytic form (Salman et al., 2019):

**Definition 4.** *Salman et al.'s Definition B.2 (**non-interactivity**) Let $\sigma : \mathbb{R}^n \to \mathbb{R}^m$ be a vector-valued non-linearity with input $\mathbf{x} \in [\boldsymbol{\ell}, \boldsymbol{u}] \subset \mathbb{R}^n$ and output $\sigma(\mathbf{x}) \in \mathbb{R}^m$. For each output $\sigma_j(\mathbf{x})$, let $I_j \subset [\![n]\!]$ be the set of $\mathbf{x}$'s entries that affect $\sigma_j(\mathbf{x})$. We call the vector-valued non-linearity non-interactive if the sets $I_j$ for $j \in [\![m]\!]$ are mutually disjoint $I_j \cap I_k = \varnothing$ for all $j \neq k \in [\![m]\!]$.*

All element-wise non-linearities such as (leaky)-ReLU, sigmoid and tanh are non-interactive. MaxPooling is also non-interactive if the stride is no smaller than the kernel size, i.e. if the receptive regions are non-overlapping.

For the ReLU non-linearity $\sigma(x) = \max(x, 0)$, the optimal convex relaxation with respect to pre-activation bounds $\ell, u$, is given by the triangle relaxation (Ehlers, 2017),

$$\underline{\sigma}(x) = \max(0, x), \quad \overline{\sigma}(x) = \frac{u}{u - \ell}(x - \ell) \tag{19}$$

see Figure 4 (middle) for an illustration.

We denote the optimal relaxation-based certification problem as $\mathcal{C}_{\text{opt}}$ and the corresponding optimal value of the objective as $p^*_{\mathcal{C}_{\text{opt}}}$.

**Optimal LP-relaxed Verification**  For piece-wise linear networks, including (leaky)-ReLU networks, the optimal single-neuron relaxation-based certification problem $\mathcal{C}_{\text{opt}}(\mathbf{c}, c_0, L, \boldsymbol{\ell}^{[\![L-1]\!]}, \boldsymbol{u}^{[\![L-1]\!]})$ is a linear programming problem and can thus be solved exactly with off-the-shelf LP solvers. Two steps are required (Salman et al., 2019): (a) We first need to obtain optimal single-neuron relaxation-based pre-activation bounds for all neurons in the network except those in the logit layer. (b) We then solve the LP-relaxed (primal or dual) certification problem exactly for the logit layer of the network.

*(a) Obtaining optimal single-neuron relaxation-based pre-activation bounds.* The optimal single-neuron relaxation-based pre-activation bounds are obtained by recursively solving the minimization $(+\mathbf{e}_j)$ resp. maximization $(-\mathbf{e}_j)$ problems $\mathcal{C}_{\text{opt}}(\pm\mathbf{e}_j, 0, i, \boldsymbol{\ell}^{[\![i-1]\!]}, \boldsymbol{u}^{[\![i-1]\!]})$ for all neurons $j \in [\![n_i]\!]$ at increasingly higher layers $i \in [\![L-1]\!]$.

*(b) Solving the LP-relaxed (primal or dual) certification problem for the logit layer.* We then solve the linear program $\mathcal{C}_{\text{opt}}(\mathbf{c}, 0, L, \boldsymbol{\ell}^{[\![L-1]\!]}, \boldsymbol{u}^{[\![L-1]\!]})$ for all $\mathbf{c} \in \{\mathbf{e}_{\hat{k}(\mathbf{x})} - \mathbf{e}_k\}_{k \neq \hat{k}(\mathbf{x})}$ with the above pre-activation bounds. If the solutions of all linear programs are positive, i.e. if $p^*_{\mathcal{C}_{\text{opt}}} > 0$ (primal) or $d^*_{\mathcal{C}_{\text{opt}}} > 0$ (dual) for all $\mathbf{c} \in \{\mathbf{e}_{\hat{k}(\mathbf{x})} - \mathbf{e}_k\}_{k \neq \hat{k}(\mathbf{x})}$, the network is certifiably robust w.r.t. $\mathbf{x}$ and $\mathcal{B}(\mathbf{x})$.

Note that the number of optimization sub-problems that need to be solved scales linearly with the number of neurons, which can easily be in the millions for deep neural networks (Salman et al., 2019). For this reason, much of the literature on certification for deep neural networks has focused on efficiently computing *approximate* pre-activation bounds. An efficient and scalable alternative to solving the optimal LP-relaxed verification problem exactly is to only solve (b) exactly and instead of (a) to greedily compute approximate but sound pre-activation bounds.

As we will see shortly, tighter pre-activation bounds also yield tighter relaxations when over-approximating the non-linearities. Most of the existing relaxation-based certification methods in use today are based on a variant of one of two greedy (pre)-activation bound computation paradigms: (i) interval bound propagation (Algorithm 1) or (ii) relaxation-based backsubstitution (Algorithm 2). The pre-activation bounds play a crucial role in these certification methods: the tighter they are, the lower the false-negative rate of the method.

**Interval Bound Propagation.**  The simplest possible method to obtain approximate but sound pre-activation bounds is given by the Interval Bound Propagation (IBP) algorithm (Dvijotham et al., 2018), which is based on the idea that valid layer-wise pre-activation bounds can be obtained by considering a separate worst-case *previous-layer perturbation* $\boldsymbol{\xi}$ for each row $\mathbf{W}^{(i)}_{j,:}$, satisfying the constraint that $\boldsymbol{\xi}$ is within the previous-layer lower- and upper- pre-activation bounds $\boldsymbol{\ell}^{(i-1)}_{j'} \leqslant \boldsymbol{\xi}_{j'} \leqslant \boldsymbol{u}^{(i-1)}_{j'}, \forall j' \in [\![n_{i-1}]\!]$. The $j'$-th perturbation entry of the greedy solution is uniquely determined by the sign of the $j'$-th entry of the vector $\mathbf{W}^{(i)}_{j,:}$. The corresponding bounds are

$$\boldsymbol{\ell}^{(i)}_j \geqslant \left[\mathbf{W}^{(i)}_{j,:}\right]_+ \sigma(\boldsymbol{\ell}^{(i-1)}) + \left[\mathbf{W}^{(i)}_{j,:}\right]_- \sigma(\boldsymbol{u}^{(i-1)}) + \mathbf{b}^{(i)}_j$$

$$\boldsymbol{u}^{(i)}_j \leqslant \left[\mathbf{W}^{(i)}_{j,:}\right]_- \sigma(\boldsymbol{\ell}^{(i-1)}) + \left[\mathbf{W}^{(i)}_{j,:}\right]_+ \sigma(\boldsymbol{u}^{(i-1)}) + \mathbf{b}^{(i)}_j \tag{20}$$

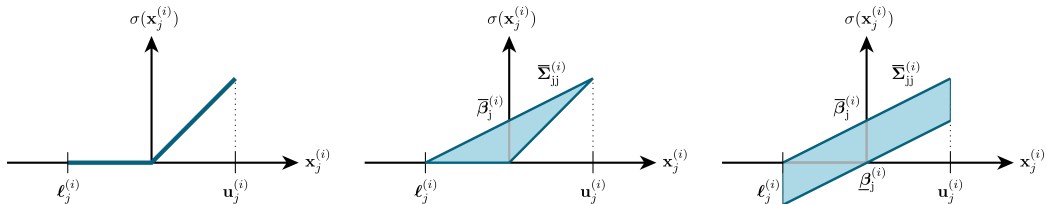

*Figure 4.* (Left) ReLU with pre-activation bounds $\boldsymbol{\ell}_j^{(i)}$ and $\boldsymbol{u}_j^{(i)}$. (Middle) Optimal ReLU relaxation. (Right) Special relaxation where $\underline{\boldsymbol{\Sigma}}^{(i)} = \overline{\boldsymbol{\Sigma}}^{(i)}$. The slopes, upper- and lower- offsets are determined by $\underline{\boldsymbol{\Sigma}}_{jj}^{(i)}, \overline{\boldsymbol{\Sigma}}_{jj}^{(i)}$ and $\underline{\boldsymbol{\beta}}_j^{(i)}, \overline{\boldsymbol{\beta}}_j^{(i)}$ respectively.

The computational complexity of the Interval Bound Propagation algorithm is linear in the number of layers $L$. The complete algorithm is shown in Algorithm 1.

---

**Algorithm 1** Interval Bound Propagation

---

**input:** Parameters $\{\mathbf{W}^{(i)}, \mathbf{b}^{(i)}\}_{i=1}^{L}$, input $\mathbf{x}$, $p$-norm, perturbation size $\epsilon$
$\boldsymbol{\ell}^{(1)} = \mathbf{W}^{(1)}\mathbf{x} - \epsilon\|\mathbf{W}^{(1)}\|_{p*..} + \mathbf{b}^{(1)}$
$\boldsymbol{u}^{(1)} = \mathbf{W}^{(1)}\mathbf{x} + \epsilon\|\mathbf{W}^{(1)}\|_{p*..} + \mathbf{b}^{(1)}$
**for** $i = 2, \ldots, L$ **do**
$\quad \boldsymbol{\ell}^{(i)} = \mathbf{W}_+^{(i)}\sigma(\boldsymbol{\ell}^{(i-1)}) + \mathbf{W}_-^{(i)}\sigma(\boldsymbol{u}^{(i-1)}) + \mathbf{b}^{(i)}$
$\quad \boldsymbol{u}^{(i)} = \mathbf{W}_-^{(i)}\sigma(\boldsymbol{\ell}^{(i-1)}) + \mathbf{W}_+^{(i)}\sigma(\boldsymbol{u}^{(i-1)}) + \mathbf{b}^{(i)}$
**end for**
**output:** bounds $\{\boldsymbol{\ell}^{(i)}, \boldsymbol{u}^{(i)}\}_{i=1}^{L}$

---

A more sophisticated class of approximate but sound pre-activation bounds can be obtained by greedy relaxation-based backsubstitution.

**Linear bounding functions**  Of particular interest is the case where each non-linear layer is bounded by exactly one *linear* lower relaxation function $\underline{\boldsymbol{\Phi}}^{(i)}(\cdot)$ and one *linear* upper relaxation function $\overline{\boldsymbol{\Phi}}^{(i)}(\cdot)$, as the corresponding relaxation-based verification problem can be solved greedily in this case (Salman et al., 2019). For ReLU non-linearities we have the following expressions for the linear bounding functions:

**Proposition 1.** *(Linear ReLU relaxation functions (Weng et al., 2018)). Each ReLU layer can be bounded as follows*

$$\underline{\boldsymbol{\Phi}}^{(i)}(\mathbf{x}^{(i)}) \leqslant \sigma(\mathbf{x}^{(i)}) \leqslant \overline{\boldsymbol{\Phi}}^{(i)}(\mathbf{x}^{(i)}) \,, \tag{21}$$

$\forall \, \boldsymbol{\ell}^{(i)} \leqslant \mathbf{x}^{(i)} \leqslant \boldsymbol{u}^{(i)}$, *with* $\underline{\boldsymbol{\Phi}}^{(i)}(\mathbf{x}^{(i)}) := \underline{\boldsymbol{\Sigma}}^{(i)}\mathbf{x}^{(i)} + \underline{\boldsymbol{\beta}}^{(i)}, \overline{\boldsymbol{\Phi}}^{(i)}(\mathbf{x}^{(i)}) := \overline{\boldsymbol{\Sigma}}^{(i)}\mathbf{x}^{(i)} + \overline{\boldsymbol{\beta}}^{(i)}$, *where the (diagonal) relaxation slope matrices* $\underline{\boldsymbol{\Sigma}}^{(i)}, \overline{\boldsymbol{\Sigma}}^{(i)}$ *and lower- and upper- offset vectors* $\underline{\boldsymbol{\beta}}^{(i)}, \overline{\boldsymbol{\beta}}^{(i)}$ *are given by*

$$\overline{\boldsymbol{\Sigma}}_{jj}^{(i)} = \begin{cases} 0 \\ 1 \\ \frac{\boldsymbol{u}_j^{(i)}}{\boldsymbol{u}_j^{(i)} - \boldsymbol{\ell}_j^{(i)}} \end{cases}, \, \underline{\boldsymbol{\Sigma}}_{jj}^{(i)} = \begin{cases} 0 \\ 1 \\ \alpha_j^{(i)} \end{cases}, \, \overline{\boldsymbol{\beta}}_j^{(i)} = \begin{cases} 0 \\ 0 \\ \frac{-\boldsymbol{u}_j^{(i)}\boldsymbol{\ell}_j^{(i)}}{\boldsymbol{u}_j^{(i)} - \boldsymbol{\ell}_j^{(i)}} \end{cases}, \, \underline{\boldsymbol{\beta}}_j^{(i)} = \begin{cases} 0 & \text{if } j \in \mathcal{I}_-^{(i)} \\ 0 & \text{, if } j \in \mathcal{I}_+^{(i)} \\ 0 & \text{if } j \in \mathcal{I}^{(i)} \end{cases} \tag{22}$$

*with* $0 \leqslant \alpha_j^{(i)} \leqslant 1$, *and where* $\mathcal{I}_-^{(i)}, \mathcal{I}_+^{(i)}$, *and* $\mathcal{I}^{(i)}$ *denote the sets of activations* $j \in [\![n_i]\!]$ *in layer* $i$ *where the lower and upper pre-activation bounds* $\boldsymbol{\ell}_j^{(i)}, \boldsymbol{u}_j^{(i)}$ *are both negative, both positive, or span zero respectively. See Figure 4 (right) for an illustration.*

Proofs can be found in (Weng et al., 2018; Zhang et al., 2018; Salman et al., 2019). See (Zhang et al., 2018) or (Liu et al., 2019) for how to define the relaxation slope matrices, lower- and upper- offset vectors for general activation functions.

Note that the linear upper bound is the optimal convex relaxation for the ReLU non-linearity, cf. Equation 19. For the lower bound, the optimal convex relaxation is not achievable as one linear function, however, and we can use any "sub-gradient" relaxation $\sigma_j(\mathbf{x}^{(i)}) = \alpha_j^{(i)}\mathbf{x}_j^{(i)}$ with $0 \leqslant \alpha_j^{(i)} \leqslant 1$, cf. Equation 19.

Choosing the same slope for the lower- and upper- relaxation functions, i.e. $\underline{\boldsymbol{\Sigma}}^{(i)} = \overline{\boldsymbol{\Sigma}}^{(i)}$, recovers Fast-Lin (Weng et al., 2018) and is equivalent to DeepZ (Singh et al., 2018). The same procedure is also used to compute pre-activation bounds in (Wong et al., 2018) (Algorithm 1). The case where the slopes $\underline{\boldsymbol{\Sigma}}^{(i)} \neq \overline{\boldsymbol{\Sigma}}^{(i)}$ of the lower relaxation function $\underline{\boldsymbol{\Phi}}^{(i)}(\cdot)$ are selected adaptively, $\alpha_j^{(i)} \in \{0, 1\}$, depending on which relaxation has the smaller volume, recovers CROWN (Zhang et al., 2018) and is equivalent to DeepPoly (Singh et al., 2019b).

**Greedy Backsubstitution**   The relaxation-based certification problem with linear bounding functions can be solved greedily in a layer-by-layer fashion (Wong & Kolter, 2018; Weng et al., 2018; Zhang et al., 2018; Salman et al., 2019). For instance, to obtain bounds on $\mathbf{x}_j^{(i+1)} = \mathbf{W}_{j\cdot\cdot}^{(i+1)}\sigma(\mathbf{x}^{(i)}) + \mathbf{b}_j^{(i+1)}$, we greedily replace the non-linearities $\sigma(\mathbf{x}^{(i)})$ with their lower and upper relaxation functions, $\underline{\boldsymbol{\Phi}}^{(i)}(\mathbf{x}^{(i)})$ resp. $\overline{\boldsymbol{\Phi}}^{(i)}(\mathbf{x}^{(i)})$, in such a way that we under-estimate the lower bounds $\boldsymbol{\ell}_j^{(i+1)}$ and over-estimate the upper bounds $\boldsymbol{u}_j^{(i+1)}$. Specifically, the bounding functions for $\sigma(\mathbf{x}^{(i)})$ are chosen based on the signs of the elements of the $j$-th row of the weight matrix, $\mathbf{W}_{j\cdot\cdot}^{(i+1)}$, i.e.

$$
\begin{aligned}
\boldsymbol{\ell}_j^{(i+1)} &\geqslant \left[\mathbf{W}_{j\cdot\cdot}^{(i+1)}\right]_+\underline{\boldsymbol{\Phi}}^{(i)}(\mathbf{x}^{(i)}) + \left[\mathbf{W}_{j\cdot\cdot}^{(i+1)}\right]_-\overline{\boldsymbol{\Phi}}^{(i)}(\mathbf{x}^{(i)}) + \mathbf{b}_j^{(i)} \\
&= \left(\left[\mathbf{W}_{j\cdot\cdot}^{(i+1)}\right]_+\underline{\boldsymbol{\Sigma}}^{(i)} + \left[\mathbf{W}_{j\cdot\cdot}^{(i+1)}\right]_-\overline{\boldsymbol{\Sigma}}^{(i)}\right)\mathbf{x}^{(i)} \\
&\quad + \left[\mathbf{W}_{j\cdot\cdot}^{(i+1)}\right]_+\underline{\boldsymbol{\beta}}^{(i)} + \left[\mathbf{W}_{j\cdot\cdot}^{(i+1)}\right]_-\overline{\boldsymbol{\beta}}^{(i)} + \mathbf{b}_j^{(i)} \\
\boldsymbol{u}_j^{(i+1)} &\leqslant \left[\mathbf{W}_{j\cdot\cdot}^{(i+1)}\right]_-\underline{\boldsymbol{\Phi}}^{(i)}(\mathbf{x}^{(i)}) + \left[\mathbf{W}_{j\cdot\cdot}^{(i+1)}\right]_+\overline{\boldsymbol{\Phi}}^{(i)}(\mathbf{x}^{(i)}) + \mathbf{b}_j^{(i)} \\
&= \left(\left[\mathbf{W}_{j\cdot\cdot}^{(i+1)}\right]_-\underline{\boldsymbol{\Sigma}}^{(i)} + \left[\mathbf{W}_{j\cdot\cdot}^{(i+1)}\right]_+\overline{\boldsymbol{\Sigma}}^{(i)}\right)\mathbf{x}^{(i)} \\
&\quad + \left[\mathbf{W}_{j\cdot\cdot}^{(i+1)}\right]_-\underline{\boldsymbol{\beta}}^{(i)} + \left[\mathbf{W}_{j\cdot\cdot}^{(i+1)}\right]_+\overline{\boldsymbol{\beta}}^{(i)} + \mathbf{b}_j^{(i)}
\end{aligned}
\tag{23}
$$

Similarly, in the expression for $\mathbf{x}^{(i)} = \mathbf{W}^{(i)}\sigma(\mathbf{x}^{(i-1)}) + \mathbf{b}^{(i)}$ in the bounds above, we can greedily replace the non-linearities $\sigma(\mathbf{x}^{(i-1)})$ with their relaxation functions $\underline{\boldsymbol{\Phi}}^{(i-1)}(\mathbf{x}^{(i-1)})$ resp. $\overline{\boldsymbol{\Phi}}^{(i-1)}(\mathbf{x}^{(i-1)})$ depending on the signs of $\left(\left[\mathbf{W}_{j\cdot\cdot}^{(i+1)}\right]_+\underline{\boldsymbol{\Sigma}}^{(i)} + \left[\mathbf{W}_{j\cdot\cdot}^{(i+1)}\right]_-\overline{\boldsymbol{\Sigma}}^{(i)}\right)\mathbf{W}^{(i)}$ and $\left(\left[\mathbf{W}_{j\cdot\cdot}^{(i+1)}\right]_-\underline{\boldsymbol{\Sigma}}^{(i)} + \left[\mathbf{W}_{j\cdot\cdot}^{(i+1)}\right]_+\overline{\boldsymbol{\Sigma}}^{(i)}\right)\mathbf{W}^{(i)}$, thus obtaining linear bounds for $\mathbf{x}_j^{(i+1)}$ in terms of $\mathbf{x}^{(i-1)}$. By the same argument we can continue this backsubstitution process until we reach the input $\mathbf{x}^{(0)}$, thus getting bounds on $\mathbf{x}_j^{(i+1)}$ in terms of $\mathbf{x}^{(0)}$ of the following form, which holds for all $\mathbf{x}^{(0)} \in \mathcal{B}(\mathbf{x})$,

$$
\underline{\boldsymbol{\Lambda}}_{j\cdot\cdot}^{(i+1)}\mathbf{x}^{(0)} + \underline{\boldsymbol{\gamma}}_j^{(i+1)} \leqslant \mathbf{x}_j^{(i+1)} \leqslant \overline{\boldsymbol{\Lambda}}_{j\cdot\cdot}^{(i+1)}\mathbf{x}^{(0)} + \overline{\boldsymbol{\gamma}}_j^{(i+1)}
\tag{24}
$$

where $\underline{\boldsymbol{\Lambda}}_{j\cdot\cdot}^{(i+1)}, \overline{\boldsymbol{\Lambda}}_{j\cdot\cdot}^{(i+1)}$ capture the products of weight matrices and relaxation slope matrices, while $\underline{\boldsymbol{\gamma}}_j^{(i+1)}, \overline{\boldsymbol{\gamma}}_j^{(i+1)}$ collect products of weight matrices, relaxation matrices and bias terms. Explicit expressions for $\underline{\boldsymbol{\Lambda}}_{j\cdot\cdot}^{(i+1)}, \overline{\boldsymbol{\Lambda}}_{j\cdot\cdot}^{(i+1)}, \underline{\boldsymbol{\gamma}}_j^{(i+1)}, \overline{\boldsymbol{\gamma}}_j^{(i+1)}$ can be found in (Weng et al., 2018; Zhang et al., 2018). The above recursion is quite remarkable, as it allows to linearly bound the output of the non-linear mapping $\mathbf{x}^{(i+1)} \equiv \mathbf{x}^{(i+1)}(\mathbf{x}^{(0)})$ for all $\mathbf{x}^{(0)} \in \mathcal{B}(\mathbf{x})$.

With the above expressions for the linear functions bounding $\mathbf{x}_j^{(i+1)}$ in terms of $\mathbf{x}^{(0)}$, we can compute lower and upper bounds $\boldsymbol{\ell}_j^{(i+1)}, \boldsymbol{u}_j^{(i+1)}$ by considering the worst-case $\mathbf{x}^{(0)} \in \mathcal{B}(\mathbf{x})$. For instance, when the uncertainty set is an $\ell_p$-norm ball $\mathcal{B}_p(\mathbf{x}; \epsilon) := \{\mathbf{x}^* : ||\mathbf{x}^* - \mathbf{x}||_p \leqslant \epsilon\}$, the bounds are given as

$$
\begin{aligned}
\boldsymbol{\ell}_j^{(i+1)} &\geqslant \underline{\boldsymbol{\Lambda}}_{j\cdot\cdot}^{(i+1)}\mathbf{x} - \epsilon||\underline{\boldsymbol{\Lambda}}_{j\cdot\cdot}^{(i+1)}||_{p*} + \underline{\boldsymbol{\gamma}}_j^{(i+1)} \\
\boldsymbol{u}_j^{(i+1)} &\leqslant \overline{\boldsymbol{\Lambda}}_{j\cdot\cdot}^{(i+1)}\mathbf{x} + \epsilon||\overline{\boldsymbol{\Lambda}}_{j\cdot\cdot}^{(i+1)}||_{p*} + \overline{\boldsymbol{\gamma}}_j^{(i+1)}
\end{aligned}
\tag{25}
$$

where $p*$ dentos the Hölder conjugate of p, given by $1/p + 1/p* = 1$.

In practice, in order to be able to replace all the non-linearities with their bounding functions from layer $i$ all the way down to the input layer, we need pre-activation bounds for all layers $i' < i$, since the expressions for the relaxation slope matrices $\underline{\boldsymbol{\Sigma}}^{(i')}, \overline{\boldsymbol{\Sigma}}^{(i')}$ and offset vectors $\underline{\boldsymbol{\beta}}^{(i')}, \overline{\boldsymbol{\beta}}^{(i')}$ depend on those pre-activation bounds. The full greedy solution thus proceeds in a layer-by-layer fashion, starting from the first layer up to the last layer, where for each layer the backsubstitution to the input is computed based on the pre-activation bounds of previous layers (computed with the same greedy approach). Hence, the computational complexity of the full greedy solution of the relaxation-based certification problem is quadratic in the number of layers $L$ of the network. The complete algorithm is shown in Section 5.4 in the Appendix.

Finally, note that while the activation over-approximation introduces looseness, relaxation-based bounds admit cancellations between positive and negative entries in weight matrices that are otherwise missed when considering the positive and negative parts of the weight matrices separately as in the Interval Bound Propagation algorithm.

**Dual formulations**   Instead of solving the certification problem $\mathcal{O}(\mathbf{c}, c_0, L, \boldsymbol{\ell}^{[\![L-1]\!]}, \boldsymbol{u}^{[\![L-1]\!]})$ in Equation 3 or the corresponding relaxation $\mathcal{C}(\mathbf{c}, c_0, L, \boldsymbol{\ell}^{[\![L-1]\!]}, \boldsymbol{u}^{[\![L-1]\!]})$ in Equation 4 in their *primal* forms, we can also solve the corresponding *dual* formulations. The Lagrangian dual of the certification problem in Equation 3 is given by (Dvijotham et al., 2018; Salman et al., 2019)

$$g_{\mathcal{O}}(\boldsymbol{\mu}^{[\![L]\!]}) = \min_{\mathbf{x}^{[\![0,L]\!]}} \ \mathbf{c}^{\top}\mathbf{x}^{(L)} + c_0 + \sum_{i=1}^{L} \boldsymbol{\mu}^{(i)\top} \left( \mathbf{x}^{(i)} - \mathbf{W}^{(i)}\sigma(\mathbf{x}^{(i-1)}) - \mathbf{b}^{(i)} \right) \tag{26}$$
$$\text{s.t.} \ \mathbf{x}^{(0)} \in \mathcal{B}(\mathbf{x}), \ \boldsymbol{\ell}^{(i)} \leqslant \mathbf{x}^{(i)} \leqslant \boldsymbol{u}^{(i)} \ \text{for} \ i \in [\![L-1]\!]$$

The Lagrangian dual of the relaxation-based certification problem in Equation 4 is given by (Wong & Kolter, 2018; Salman et al., 2019)

$$\begin{aligned} &g_{\mathcal{C}}(\boldsymbol{\mu}^{[\![L]\!]}, \underline{\boldsymbol{\lambda}}^{[\![0,L-1]\!]}, \overline{\boldsymbol{\lambda}}^{[\![0,L-1]\!]}) \\ &= \min_{\mathbf{x}^{[\![0,L]\!]}, \mathbf{z}^{[\![0,L-1]\!]}} \ \mathbf{c}^{\top}\mathbf{x}^{(L)} + c_0 + \sum_{i=1}^{L} \boldsymbol{\mu}^{(i)\top} \left( \mathbf{x}^{(i)} - \mathbf{W}^{(i)}\mathbf{z}^{(i-1)} - \mathbf{b}^{(i)} \right) \\ &\quad - \sum_{i=0}^{L-1} \underline{\boldsymbol{\lambda}}^{(i)\top} \left( \mathbf{z}^{(i)} - \underline{\sigma}(\mathbf{x}^{(i)}) \right) + \sum_{i=0}^{L-1} \overline{\boldsymbol{\lambda}}^{(i)\top} \left( \mathbf{z}^{(i)} - \overline{\sigma}(\mathbf{x}^{(i)}) \right) \end{aligned} \tag{27}$$
$$\text{s.t.} \ \mathbf{x}^{(0)} \in \mathcal{B}(\mathbf{x}), \ \boldsymbol{\ell}^{(i)} \leqslant \mathbf{x}^{(i)} \leqslant \boldsymbol{u}^{(i)} \ \text{for} \ i \in [\![L-1]\!]$$

where $\mathbf{z}^{(i)}$ represent the post-activation variables.

By weak duality (Boyd et al., 2004), we have for the original dual

$$d_{\mathcal{O}}^* := \max_{\boldsymbol{\mu}^{[\![L]\!]}} g_{\mathcal{O}}(\boldsymbol{\mu}^{[\![L]\!]}) \leqslant p_{\mathcal{O}}^* \tag{28}$$

respectively for the convex-relaxation based dual

$$d_{\mathcal{C}}^* := \max_{\boldsymbol{\mu}^{[\![L]\!]}, \underline{\boldsymbol{\lambda}}^{[\![0,L-1]\!]} \geqslant 0, \overline{\boldsymbol{\lambda}}^{[\![0,L-1]\!]} \geqslant 0} g_{\mathcal{C}}(\boldsymbol{\mu}^{[\![L]\!]}, \underline{\boldsymbol{\lambda}}^{[\![0,L-1]\!]}, \overline{\boldsymbol{\lambda}}^{[\![0,L-1]\!]}) \leqslant p_{\mathcal{C}}^* \tag{29}$$

Hence, if $d_{\mathcal{O}}^* > 0$ resp. $d_{\mathcal{C}}^* > 0$ for all $\mathbf{c} \in \{\mathbf{e}_{\hat{k}(\mathbf{x})} - \mathbf{e}_k\}_{k \neq \hat{k}(\mathbf{x})}$, with $c_0 = 0$ and valid pre-activation bounds $\boldsymbol{\ell}^{[\![L-1]\!]}, \boldsymbol{u}^{[\![L-1]\!]}$, the network is certifiably robust with respect to $\mathbf{x}$ and $\mathcal{B}(\mathbf{x})$.

In fact, for the convex-relaxation based certification problem, one can show that strong duality ($d_{\mathcal{C}}^* = p_{\mathcal{C}}^*$) holds under relatively mild conditions (finite Lipschitz constant for the bounding functions $\underline{\sigma}(\cdot), \overline{\sigma}(\cdot)$), see Theorem 4.1 in (Salman et al., 2019). Moreover, one can even show that the dual of the optimal single-neuron convex relaxation based certification problem is equivalent to the dual of the original certification problem, i.e. $d_{\mathcal{C}_{\text{opt}}}^* = d_{\mathcal{O}}^*$, see Theorem 4.2 in (Salman et al., 2019).

Despite these equivalences, there are still good reasons to solve the dual instead of the primal problem. (Salman et al., 2019) recommend solving the dual problem because (i) the dual problem can be formulated as an unconstrained optimization problem, whereas the primal is a constrained optimization problem and (ii) the dual optimization process can be stopped anytime to give a valid lower bound on $p_{\mathcal{O}}^*$ (thanks to weak duality).

## 5.4. Single-Neuron Relaxation-based Bound Computation in the Special Case $\underline{\Sigma}^{(i)} = \overline{\Sigma}^{(i)} \equiv \Sigma^{(i)}$

When each non-linear layer is bounded by exactly one linear lower relaxation function and one linear upper relaxation function, the relaxed verification problem can be solved greedily (Salman et al., 2019).

In the special case $\underline{\Sigma}^{(i)} = \overline{\Sigma}^{(i)} \equiv \Sigma^{(i)}$, we can derive concise closed-form expressions for the lower- and upper pre-activation bounds. To this end, we replace the activations with their enveloping relaxation functions $\underline{\Phi}^{(i)}(\mathbf{x}^{(i)}) = \Sigma^{(i)}\mathbf{x}^{(i)} + \underline{\beta}^{(i)}$ resp. $\overline{\Phi}^{(i)}(\mathbf{x}^{(i)}) = \Sigma^{(i)}\mathbf{x}^{(i)} + \overline{\beta}^{(i)}$ where the offsets $\underline{\beta}_{j'}^{(i)}, \overline{\beta}_{j'}^{(i)}$ are for each layer $i$ and unit $j'$ chosen such that we under-estimate lower bounds $\ell_j^{(i)}$ and over-estimate upper bounds $u_j^{(i)}$,

$$
\begin{aligned}
\ell_j^{(i)} \geqslant &\min_{\xi \in \mathcal{B}(\mathbf{0})} \min_{\Phi_{j'}^{(i-1)} \in \{\underline{\Phi}_{j'}^{(i-1)}, \overline{\Phi}_{j'}^{(i-1)}\}} \cdots \min_{\Phi_{j'}^{(1)} \in \{\underline{\Phi}_{j'}^{(1)}, \overline{\Phi}_{j'}^{(1)}\}} \Big\{ \\
&\mathbf{W}_{j\cdot\cdot}^{(i)} \Phi^{(i-1)}(\cdots \mathbf{W}^{(2)}\Phi^{(1)}(\mathbf{W}^{(1)}(\mathbf{x}+\xi)+\mathbf{b}^{(1)})+\mathbf{b}^{(2)}\cdots)+\mathbf{b}^{(L)} \Big\} \\
= &\min_{\xi \in \mathcal{B}(\mathbf{0})} \min_{\beta_{j'}^{(i-1)} \in \{\underline{\beta}_{j'}^{(i-1)}, \overline{\beta}_{j'}^{(i-1)}\}} \cdots \min_{\beta_{j'}^{(1)} \in \{\underline{\beta}_{j'}^{(1)}, \overline{\beta}_{j'}^{(1)}\}} \Big\{ \mathbf{W}_{j\cdot\cdot}^{(i)}(\Sigma^{(i-1)}(\cdots \mathbf{W}^{(2)}( \\
&\Sigma^{(1)}(\mathbf{W}^{(1)}(\mathbf{x}+\xi)+\mathbf{b}^{(1)})+\beta^{(1)})+\mathbf{b}^{(2)}\cdots)+\beta^{(i-1)})+\mathbf{b}_j^{(i)} \Big\} \\
u_j^{(i)} \leqslant &\max_{\xi \in \mathcal{B}(\mathbf{0})} \max_{\Phi_{j'}^{(i-1)} \in \{\underline{\Phi}_{j'}^{(i-1)}, \overline{\Phi}_{j'}^{(i-1)}\}} \cdots \max_{\Phi_{j'}^{(1)} \in \{\underline{\Phi}_{j'}^{(1)}, \overline{\Phi}_{j'}^{(1)}\}} \Big\{ \\
&\mathbf{W}_{j\cdot\cdot}^{(i)} \Phi^{(i-1)}(\cdots \mathbf{W}^{(2)}\Phi^{(1)}(\mathbf{W}^{(1)}(\mathbf{x}+\xi)+\mathbf{b}^{(1)})+\mathbf{b}^{(2)}\cdots)+\mathbf{b}^{(L)} \Big\} \\
= &\max_{\xi \in \mathcal{B}(\mathbf{0})} \max_{\beta_{j'}^{(i-1)} \in \{\underline{\beta}_{j'}^{(i-1)}, \overline{\beta}_{j'}^{(i-1)}\}} \cdots \max_{\beta_{j'}^{(1)} \in \{\underline{\beta}_{j'}^{(1)}, \overline{\beta}_{j'}^{(1)}\}} \Big\{ \mathbf{W}_{j\cdot\cdot}^{(i)}(\Sigma^{(i-1)}(\cdots \mathbf{W}^{(2)}( \\
&\Sigma^{(1)}(\mathbf{W}^{(1)}(\mathbf{x}+\xi)+\mathbf{b}^{(1)})+\beta^{(1)})+\mathbf{b}^{(2)}\cdots)+\beta^{(i-1)})+\mathbf{b}_j^{(i)} \Big\}
\end{aligned}
\tag{30}
$$

Note that the $j'$ span different ranges for the different variables.

For the sake of simplicity, we introduce special notation for products over decreasing index sequences, where the index is counted down from the product-superscript to the product-subscript

**Definition 5.** *(Products over decreasing index sequences). Let $\mathbf{A}^{(k)}, k \in [m, \ldots, n]$ be an arbitrary sequence of (real-valued) matrices with matching inner dimensions. For ease of notation, define*

$$
\prod_{m}^{k=n} \mathbf{A}^{(k)} = \prod_{k=0}^{n-m} \mathbf{A}^{(n-k)} = \mathbf{A}^{(n)}\mathbf{A}^{(n-1)}\cdots\mathbf{A}^{(m)} \quad \text{where} \quad n \geqslant m
\tag{31}
$$

We use the usual convention that the empty product equals one, i.e. $\prod_{k \in \varnothing}(\cdot) = 1$. Thus, $\prod_{k=m}^{n}(\cdot) = 1$ and $\prod_m^{k=n}(\cdot) = 1$ whenever $n < m$.

We also introduce the following closed-form expressions for the layer-wise activation-relaxation

**Definition 6.** *(Closed-form expressions for layer-wise activation-relaxation).*

$$
\phi^{(i)} = \Big( \prod_{2}^{k=i} \mathbf{W}^{(k)}\Sigma^{(k-1)} \Big)\mathbf{W}^{(1)}\mathbf{x} + \sum_{j=1}^{i} \Big( \prod_{j+1}^{k=i} \mathbf{W}^{(k)}\Sigma^{(k-1)} \Big)\mathbf{b}^{(j)}
\tag{32}
$$

$$
\Lambda^{(j)} = \Big( \prod_{j+1}^{k=i} \mathbf{W}^{(k)}\Sigma^{(k-1)} \Big)\mathbf{W}^{(j)} \quad \text{for} \quad j = 1, \ldots, i
\tag{33}
$$

Rewriting the above equations using the explicit expressions for the concatenation of layers,

$$
\ell_j^{(i)} \geqslant \min_{\boldsymbol{\xi} \in \mathcal{B}(\mathbf{0})} \min_{\boldsymbol{\beta}_{j'}^{(i-1)} \in \{\underline{\boldsymbol{\beta}}_{j'}^{(i-1)}, \overline{\boldsymbol{\beta}}_{j'}^{(i-1)}\}} \cdots \min_{\boldsymbol{\beta}_{j'}^{(1)} \in \{\underline{\boldsymbol{\beta}}_{j'}^{(1)}, \overline{\boldsymbol{\beta}}_{j'}^{(1)}\}} \left\{ \phi_j^{(i)} + \boldsymbol{\Lambda}_{j..}^{(1)} \boldsymbol{\xi} + \sum_{k=2}^{i} \boldsymbol{\Lambda}_{j..}^{(k)} \boldsymbol{\beta}^{(k-1)} \right\}
$$

$$
\boldsymbol{u}_j^{(i)} \leqslant \max_{\boldsymbol{\xi} \in \mathcal{B}(\mathbf{0})} \max_{\boldsymbol{\beta}_{j'}^{(i-1)} \in \{\underline{\boldsymbol{\beta}}_{j'}^{(i-1)}, \overline{\boldsymbol{\beta}}_{j'}^{(i-1)}\}} \cdots \max_{\boldsymbol{\beta}_{j'}^{(1)} \in \{\underline{\boldsymbol{\beta}}_{j'}^{(1)}, \overline{\boldsymbol{\beta}}_{j'}^{(1)}\}} \left\{ \phi_j^{(i)} + \boldsymbol{\Lambda}_{j..}^{(1)} \boldsymbol{\xi} + \sum_{k=2}^{i} \boldsymbol{\Lambda}_{j..}^{(k)} \boldsymbol{\beta}^{(k-1)} \right\}
$$

(34)

we can see that the different minimizations and maximizations over optimal perturbations, lower- and upper- offset vectors decouple, hence we can resolve them separately, e.g. for $\mathcal{B}_p^\epsilon(\mathbf{0})$:

$$
\boldsymbol{\ell}^{(i)} \geqslant \boldsymbol{\phi}^{(i)} - \epsilon \|\boldsymbol{\Lambda}^{(1)}\|_{p*..} + \sum_{j=2}^{i} \left( \boldsymbol{\Lambda}_+^{(j)} \underline{\boldsymbol{\beta}}^{(j-1)} + \boldsymbol{\Lambda}_-^{(j)} \overline{\boldsymbol{\beta}}^{(j-1)} \right)
$$

$$
\boldsymbol{u}^{(i)} \leqslant \boldsymbol{\phi}^{(i)} + \epsilon \|\boldsymbol{\Lambda}^{(1)}\|_{p*..} + \sum_{j=2}^{i} \left( \boldsymbol{\Lambda}_-^{(j)} \underline{\boldsymbol{\beta}}^{(j-1)} + \boldsymbol{\Lambda}_+^{(j)} \overline{\boldsymbol{\beta}}^{(j-1)} \right)
$$

(35)

where $p*$ is the Hölder conjugate $1/p + 1/p* = 1$ and where $\| \cdot \|_{p*..}$ denotes the row-wise $p*$-norm.

Choosing the same slope for the lower- and upper- relaxation functions $\alpha_j^{(i)} = \boldsymbol{u}_j^{(i)} / (\boldsymbol{u}_j^{(i)} - \boldsymbol{\ell}_j^{(i)})$, i.e. $\underline{\boldsymbol{\Sigma}}^{(i)} = \overline{\boldsymbol{\Sigma}}^{(i)}$, recovers Fast-Lin (Weng et al., 2018), while adaptively setting $\alpha_j^{(i)}$ to its boundary values depending on which relaxation has the smaller volume recovers CROWN (Zhang et al., 2018).

---

**Algorithm 2** Relaxation based Bound Computation (for $\underline{\boldsymbol{\Sigma}}^{(i)} \equiv \overline{\boldsymbol{\Sigma}}^{(i)}, \underline{\boldsymbol{\beta}}^{(i)} \equiv \mathbf{0}$)

---

**input:** Parameters $\{\mathbf{W}^{(i)}, \mathbf{b}^{(i)}\}_{i=1}^{L}$, input $\mathbf{x}$, $p$-norm, perturbation size $\epsilon$
$\boldsymbol{\Lambda}^{(1)} = \mathbf{W}^{(1)}$
$\boldsymbol{\phi}^{(1)} = \mathbf{W}^{(1)} \mathbf{x} + \mathbf{b}^{(1)}$
$\boldsymbol{\ell}^{(1)} = \boldsymbol{\phi}^{(1)} - \epsilon \|\boldsymbol{\Lambda}^{(1)}\|_{p*..}$
$\boldsymbol{u}^{(1)} = \boldsymbol{\phi}^{(1)} + \epsilon \|\boldsymbol{\Lambda}^{(1)}\|_{p*..}$
**for** $i = 2, \ldots, L$ **do**
$\quad$ Compute $\boldsymbol{\Sigma}^{(i-1)}, \overline{\boldsymbol{\beta}}^{(i-1)}$ from $\boldsymbol{\ell}^{(i-1)}, \boldsymbol{u}^{(i-1)}$
$\quad \boldsymbol{\Lambda}^{(j)} = \mathbf{W}^{(i)} \boldsymbol{\Sigma}^{(i-1)} \boldsymbol{\Lambda}^{(j)}$ for $j = 1, \ldots, i-1$
$\quad \boldsymbol{\Lambda}^{(i)} = \mathbf{W}^{(i)}$
$\quad \boldsymbol{\phi}^{(i)} = \mathbf{W}^{(i)} \boldsymbol{\Sigma}^{(i-1)} \boldsymbol{\phi}^{(i-1)} + \mathbf{b}^{(i)}$
$\quad \boldsymbol{\ell}^{(i)} = \boldsymbol{\phi}^{(i)} - \epsilon \|\boldsymbol{\Lambda}^{(1)}\|_{p*..} + \sum_{j=2}^{i} \left( \boldsymbol{\Lambda}_-^{(j)} \overline{\boldsymbol{\beta}}^{(j-1)} \right)$
$\quad \boldsymbol{u}^{(i)} = \boldsymbol{\phi}^{(i)} + \epsilon \|\boldsymbol{\Lambda}^{(1)}\|_{p*..} + \sum_{j=2}^{i} \left( \boldsymbol{\Lambda}_+^{(j)} \overline{\boldsymbol{\beta}}^{(j-1)} \right)$
**end for**
**output:** bounds $\{\boldsymbol{\ell}^{(i)}, \boldsymbol{u}^{(i)}\}_{i=1}^{L}$

---