# OpenReview forum: "A Primer on Multi-Neuron Relaxation-based Adversarial Robustness Certification"
_ICML.cc/2021/Workshop/AML — ICML 2021 Workshop AML Poster_

### Official Review · Reviewer_Kt1R · 2021-06-19
**The paper developed a unified mathematical framework  to describe relaxation-based robustness certification methods and show  how “k-ReLU” multi-neuron relaxation framework obtains tighter correlation-aware activation bounds.**

**Rating:** Accept
**Confidence:** 3

**Review:**

The paper focused on how to evaluate the robustness of defense models against any adversarial attacks and mainly used Relaxation-based Certification method to analyze. The paper proved that “k-ReLU” framework obtains tighter correlation-aware activation bounds by leveraging additional relational constraints among groups of neurons. However, it’s better to control the length of the text within four pages based on the requirements of AML. What’s more, it’s better to explain clearly the meaning of every notation when it first appeared, for example, in the left part of line 98, $e_k$ first appeared without explanation.

---

### Decision · Program_Chairs · 2021-06-21

**Decision:**

Accept (Poster)

**Comment:**

The paper proved that “k-ReLU” framework obtains tighter correlation-aware activation bounds by leveraging additional relational constraints among groups of neurons. The writing could be further improved considering the reviewer's comments.